# The Combined Partial Knockdown of *CBS* and *MPST* Genes Induces Inflammation, Impairs Adipocyte Function-Related Gene Expression and Disrupts Protein Persulfidation in Human Adipocytes

**DOI:** 10.3390/antiox11061095

**Published:** 2022-05-31

**Authors:** Jessica Latorre, Angeles Aroca, José Manuel Fernández-Real, Luis C. Romero, José María Moreno-Navarrete

**Affiliations:** 1Department of Diabetes, Endocrinology and Nutrition, Institut d’Investigació Biomèdica de Girona (IdIBGi), 17190 Salt, Spain; jlatorre@idibgi.org (J.L.); jmfreal@idibgi.org (J.M.F.-R.); 2CIBER Fisiopatología de la Obesidad y Nutrición (CIBERobn, CB06/03/010), Instituto de Salud Carlos III, 28029 Madrid, Spain; 3Instituto de Bioquímica Vegetal y Fotosíntesis, Consejo Superior de Investigaciones and Universidad de Sevilla, 41092 Seville, Spain; aaroca@us.es (A.A.); lromero@ibvf.csic.es (L.C.R.); 4Department of Medicine, Universitat de Girona, 17003 Girona, Spain

**Keywords:** human adipocytes, adipogenesis, inflammation, protein persulfidation

## Abstract

Recent studies in mice and humans demonstrated the relevance of H_2_S synthesising enzymes, such as CTH, CBS, and MPST, in the physiology of adipose tissue and the differentiation of preadipocyte into adipocytes. Here, our objective was to investigate the combined role of CTH, CBS, and MPST in the preservation of adipocyte protein persulfidation and adipogenesis. Combined partial *CTH*, *CBS*, and *MPST* gene knockdown was achieved treating fully human adipocytes with siRNAs against these transcripts (siRNA_MIX). Adipocyte protein persulfidation was analyzed using label-free quantitative mass spectrometry coupled with a dimedone-switch method for protein labeling and purification. Proteomic analysis quantified 216 proteins with statistically different levels of persulfidation in KD cells compared to control adipocytes. In fully differentiated adipocytes, *CBS* and *MPST* mRNA and protein levels were abundant, while *CTH* expression was very low. It is noteworthy that siRNA_MIX administration resulted in a significant decrease in *CBS* and *MPST* expression, without impacting on CTH. The combined partial knockdown of the *CBS* and *MPST* genes resulted in reduced cellular sulfide levels in parallel to decreased expression of relevant genes for adipocyte biology, including adipogenesis, mitochondrial biogenesis, and lipogenesis, but increased proinflammatory- and senescence-related genes. It should be noted that the combined partial knockdown of *CBS* and *MPST* genes also led to a significant disruption in the persulfidation pattern of the adipocyte proteins. Although among the less persulfidated proteins, we identified several relevant proteins for adipocyte adipogenesis and function, among the most persulfidated, key mediators of adipocyte inflammation and dysfunction as well as some proteins that might play a positive role in adipogenesis were found. In conclusion, the current study indicates that the combined partial elimination of CBS and MPST (but not CTH) in adipocytes affects the expression of genes related to the maintenance of adipocyte function and promotes inflammation, possibly by altering the pattern of protein persulfidation in these cells, suggesting that these enzymes were required for the functional maintenance of adipocytes.

## 1. Introduction

Functional adipocytes are characterized by an increased capacity to store excess fuel and produce beneficial adipokines, increased expression of adipogenic genes, but decreased expression of proinflammatory- and cellular senescence-related genes [1,2,3,4]. Optimal adipocyte function is required to maintain adipose tissue physiology and prevents obesity-associated metabolic disturbances [3,4,5,6].

Recent studies in mice and humans demonstrated the relevance of H_2_S-synthesising enzymes, such as CTH, CBS, and MPST, to preserve adipogenesis of adipose tissue, healthy fat mass expansion and insulin action [7,8]. These enzymes play a crucial role in the differentiation of adipocytes. Specifically, gene knockdown experiments in mouse 3T3-L1 cells and human preadipocytes demonstrated the relevance of CTH, CBS and MPST in adipogenesis [7,8,9]. Interestingly, increased persulfidation in relevant adipogenic proteins has also been reported in human adipocytes [8]. Since persulfidation often increases the reactivity of target proteins [10], increased persulfidation in adipogenic enzymes suggests a possible role of these post-translational modifications in the promotion of adipocyte differentiation and in the maintenance of adipogenic status. Despite these intriguing studies, to the best of our knowledge, the combined role of these enzymes in the maintenance of adipocyte function has not been yet examined.

In the present study, our objective was to investigate the combined role of CTH, CBS, and MPST in the preservation of adipocyte adipogenesis and function and to investigate how the combined partial knockdown of these enzymes might impact the persulfidation of the adipocyte proteins.

## 2. Materials and Methods

### 2.1. Human Preadipocyte Differentiation and Adipocyte Maintenance

Primary human subcutaneous preadipocytes from a non-diabetic Caucasian female with BMI < 30 kg/m^2^ (Zen-Bio Inc., Research Triangle Park, Durham, NC, USA) were cultured with Preadipocyte Medium (PM-1, Zen-Bio Inc.) in a humidified incubator at 37 °C with 5% CO_2_. Twenty-four hours after plating, cells were checked for confluence. Once reached, differentiation started with commercially available Differentiation Medium (DM-2, Zen-Bio Inc.) and Maintenance Medium (AM-1 Zen-Bio Inc.), following the manufacturer’s instructions. Two weeks after initiating differentiation, differentiated cells appeared rounded with large lipid droplets apparent in the cytoplasm. The cells were then considered mature adipocytes. Forward siRNA transfections using siRNA_MIX, which included siRNAs against *MPST*, *CBS*, and *CTH* gene expression, were performed at the end of the differentiation process.

### 2.2. siRNA-Induced Gene Knockdown in Fully Differentiated Adipocytes

Primary human subcutaneous adipocytes were forward transfected with siRNAs at the end of differentiation (on day 11). Briefly, siRNA (Sigma-Aldrich, St. Louis, MO, USA) against CTH (NM_001190463), CBS (NM_000071), and MPST (NM_001130517) and Lipofectamine RNAiMAX (Life Technologies, Darmstadt, Germany) were diluted separately with Opti-MEM I Reduced Serum Medium (Life Technologies, Darmstadt, Germany) and mixed by pipetting afterwards. The siRNA-RNAiMAX complexes were left to incubate for 20 min at room temperature and then added to the top of the adherent cells drop-wise. The final concentrations of Lipofectamine RNAiMAX and siRNAs were 1.6 μL/cm^2^ and 100 nM, respectively, in 12-well cell culture plates, and the final amount of medium per well was 1 mL. The transfection conditions included combined silencing of CTH, CBS and MSPT (100 nM), and a non-targeting siRNA (100 nM) for the vehicle. Adipocytes were harvested on day 14 of differentiation, i.e., 72 h after transfection without changing cell culture medium. Transfection efficiency was assessed by real-time PCR and Western blot. The MISSION^®^ siRNAs (Sigma-Aldrich) used were MISSION^®^ esiRNA targeting human MPST (EHU086511), CBS (EHU099041), and CTH (EHU078251). The MISSION^®^ siRNA Universal Negative Control #1 (Sigma-Aldrich, SIC001) was used as a control in all experiments (siRNA_SCR).

### 2.3. Gene Expression

RNA was extracted from cells using the RNeasy Lipid Tissue Mini Kit (Qiagen, Germantown, MD, USA). Total RNA was quantified using a spectrophotometer (GeneQuant; GE Health Care, Waukesha, WI, USA). RNA was reverse transcribed to cDNA using the High-Capacity cDNA Reverse Transcription Kit (Life Technology, Darmstadt, Germany) according to standard procedures. Commercially available Taq-Man primer and probe sets were used for gene expression. Expression was assessed by real-time PCR using the LightCycler 480 Real-Time PCR System (Roche Diagnostics, Barcelona, Spain).

The commercially available and pre-validated TaqMan^®^ primer/probe sets used were as follows: Peptidylprolyl isomerase A (cyclophilin A) (4333763, *PPIA* as endogenous control), cystathionine γ-lyase (*CTH*, Hs00542284_m1), cystathionine β-synthase (*CBS*, Hs00163925_m1), mercaptopyruvate sulfurtransferase (*MPST*, Hs00560401_m1), adiponectin (*ADIPOQ*, Hs00605917_m1), fatty acid binding protein 4, adipocyte (*FABP4*, Hs01086177_m1), peroxisome proliferator-activated receptor gamma (*PPARG*, Hs00234592_m1), perilipin 1 (*PLIN1*, Hs00160173_m1), PPARG coactivator 1 alpha (*PPARGC1A*, Hs00173304_m1), fatty acid synthase (*FASN*, Hs00188012_m1), acetyl-CoA carboxylase alpha (*ACACA*, Hs00167385_m1), solute carrier family 2 member 4 (*SLC2A4* or *GLUT4*, Hs00168966_m1), interleukin 6 (interferon, beta 2) (*IL6*, Hs00985639_m1), tumor necrosis factor (*TNF*, Hs00174128_m1), C-X-C motif chemokine ligand 8 (*CXCL8* or *IL8*, Hs00174103_m1), C–C motif chemokine ligand 2 (*CCL2* or *MCP1*, Hs00234140_m1), and tumor protein p53 (*TP53*, Hs01034249_m1).

### 2.4. Western Blot

The proteins were extracted from cells using radioimmunoprecipitation assay (RIPA) buffer (0.1% SDS, 0.5% sodium deoxy-cholate, 1% Nonidet P-40, 150 mmol/L NaCl, and 50 mmol/L Tris-HCl, pH 8.0) supplemented with a commercially available cocktail of protease inhibitors (Millipore, Sigma, Madrid, Spain). Cellular debris and lipids were eliminated by centrifugation of the solubilized samples at 12,000× *g* for 10 min at 4 °C, recovering the soluble fraction. The protein concentration was determined using the RC/DC Protein Assay (Bio-Rad Laboratories, Hercules, CA, USA). Protein extracts were processed in SDS-PAGE and transferred to nitrocellulose membranes by conventional procedures. After blocking with 5% BSA in TBS-Tween, the membranes were incubated with primary antibodies, followed by incubation with horseradish peroxidase conjugated goat anti-rabbit antibody (Cell Signaling, 7074). Protein signal was detected by chemiluminescence ChemiDoc™ Gel Imaging System (Bio-Rad, California, USA). Actin was used as a housekeeping gene-coded protein. All primary antibodies were used at 1:1000 dilutions and were the following: CBS (sc-133154, Santa Cruz Biotechnology, Santa Cruz, CA, USA), CTH (sc-365382, Santa Cruz Biotechnology), MPST (sc-376168, Santa Cruz Biotechnology), FAS (C20G5, Cell Signaling), and β-actin (sc-47778, Santa Cruz Biotechnology).

### 2.5. Quantification of Hydrogen Sulfide in Biological Matrices by LC-MS/MS

Hydrogen sulfide was quantified following a previously described method [11]. A 50 μL sample (cell lysate) was spiked with 70 μL of MBB solution (20 mM). The mixture was vortexed, incubated for 60 min at room temperature, and the derivatization reaction stopped by adding 10 μL 20% formic acid. The mixture was subjected to centrifugation at 12,000 × rpm for 10 min. The derivative sulfide dibimane was analyzed by LC-MS/MS using the ExionLC™ UPLC system (Sciex, Framingham, MA, USA) with a reversed-phase column (100 mm × 4.6 mm × 100 Å particle, Kinetex XB-C18, Phenomenex, Torrance, CA, USA). The mobile phases were 0.1% formic acid in H_2_O (Buffer A, HPLC/LCMS grade) and 0.1% formic acid in acetonitrile (Buffer B, HPLC/LC-MS grade). A 5 µL sample was loaded per injection, and the gradient, applied at a flow rate of 600 µL min^−1^, was as follows: 1.7 min 80% A, from 1.7 to 3 min linear gradient from 80% A to 20% A, 1 min isocratic 20% A, 1 min linear gradient from 20% A to 80% A, and hold 3 min at 80% A to re-equilibrate the column. Mass spectra were acquired using a QTRAP 6500 + triple quadrupole (Sciex, Framingham, MA, USA) equipped with an electrospray ionization source operating in negative ionization mode using an ion spray voltage of −4500 V. The other ESI parameters were as follows: curtain gas, 35 psi; collision gas, medium; temperature, 500 °C; nebulizer gas (GS1), 60 psi; and heater gas (GS2), 60 psi. Data were acquired with Analyst^®^ 1.7 software in the multiple reaction monitoring (MRM) mode with a detection window of 60 s. The measured ionization adducts [M-H]^−^ selected for identification and quantification were 412.9 m/z (Q1) and 190.9 m/z (Q3) and optimized declustering potential (DP − 45 V) and collision energy (CE − 24 V). Data were processed with Sciex OS^®^ software (Sciex, Redwood City, CA, USA) for peak integration and quantification.

### 2.6. Protein Persulfidation

A total of 500 µg of proteins from adipocyte lysates in 50 mM Tris-HCl, pH 8, supplemented with 1% protease inhibitor (cOmplete™, SigmaAldrich) and 2% SDS was incubated with 5 mM 4-chloro-7-nitrobenzofurazan (Cl-NBF) at 37 °C for 30 min, protected from light. A methanol/chloroform precipitation was performed to eliminate excess Cl-NBF, and protein pellets obtained were washed with cold methanol, dried, and dissolved in 50 mM Tris-HCl, pH 8, with 2% SDS, as previously described [12,13]. The proteins were then incubated with 100 µM DCP-Bio1 at 37 °C for 1.5 h. Afterwards, proteins were precipitated with methanol/chloroform and dissolved in 50 mM Tris-HCl, pH 8, supplemented with 0.1% SDS. Proteins were incubated with Sera-Mag™ Magnetic Streptavidin (Cytiva, Madrid, Spain) at 4 °C overnight with agitation and then, beads were washed with 8 column volumes of 50 mM Tris-HCl, pH 8, with 0.001% Tween, 2 column volumes of 50 mM Tris-HCl, pH 8, and 1 column volume of pure HPLC-quality water. After washing, beads were incubated 5 h with agitation with 2.25 M ammonium hydroxide at RT. The sample was then neutralized with formic acid and the protein concentration was determined. A total of 50 µg of proteins were trypsinized (GOLD, Promega) following the filter-aided sample preparation (FASP) method [14]. Trypsin was added to a trypsin:protein ratio of 1:20, and the mixture was incubated overnight at 37 °C, dried out in a RVC2 25 speedvac concentrator (Christ), and resuspended in 0.1% FA. Peptides were desalted and resuspended in 0.1% FA using C18 stage tips (Millipore). The samples were analyzed in a timsTOF Pro with PASEF (Bruker Daltonics, Billerica, MA, USA) coupled online to an Evosep ONE liquid chromatograph (Evosep). Hence, 200 ng were directly loaded onto the Evosep ONE and resolved using the 30 samples-per-day protocol. The processed data were analyzed with the MQ (Max Quant) search engine and the label-free quantification was performed using PEAKS Studio (BSI, Mississauga, Canada) [15]. Protein identification and quantification were carried out using PEAKS X software (Bioinformatics solutions). The searches were carried out against a database consisting of *Homo sapiens* (Uniprot/Swissprot), with precursor and fragment tolerances of 20 ppm and 0.05 Da. Only proteins identified with at least two peptides at FDR < 1% were considered for further analysis. Protein abundances inferred from PEAKS were loaded onto Perseus computational platform [16], log2 transformed, and imputated. A *t*-test was used to address significant differences in protein abundances within each sample group under analysis.

The mass spectrometry proteomics data have been deposited in the ProteomeXchange Consortium [17] via the PRIDE partner repository with the dataset identifier PXD032370 and 10.6019/PXD032370.

Functional protein association networks were explored using the STRING database (STRING: functional protein association networks (string-db.org) access date 26 February 2022).

### 2.7. Statistical Analysis

Statistical analyses were performed using SPSS 12.0 software. The unpaired *t*-test and nonparametric test (Mann–Whitney test) were used to analyze in vitro experimental data. The levels of statistical significance were set at *p* < 0.05.

## 3. Results and Discussion

### 3.1. The Relevance of CTH, CBS and MPST on Human Adipocytes

In fully differentiated adipocytes, *CBS* and *MPST* mRNA and protein levels were abundant, whereas CTH expression was very low (Figure 1A,B). It should be noted that siRNA_MIX administration resulted in a significant decrease in *CBS* and *MPST* expression, without impacting on CTH (Figure 1A,B). Possibly, the very low transcription and turnover of CTH make it resistant to the effect of siRNA. The combined partial knockdown of *CBS* and *MPST* genes led to a significant reduction in cellular sulfide levels (Figure 1C), indicating that H_2_S biosynthesis was attenuated in these cells. Interestingly, the combined partial *CBS* and *MPST* gene knockdown also resulted in decreased expression of relevant genes for adipocyte biology, including adipogenesis (*ADIPOQ, FABP4, PLIN1, SLC2A4, PPARG*), mitochondrial biogenesis (*PPARGC1A*), and lipogenesis (*ACACA* and *FASN* mRNA, and FAS protein) (Figure 1D–L), but increased proinflammatory- and senescence-related genes (*TP53*) (Figure 1M–Q). Importantly, these data indicated that MPST and CBS were not only required during adipocyte differentiation of mouse 3T3-L1 cells [9] or human preadipocytes [8], but these proteins were also necessary for the functional maintenance of adipocytes. In contrast to that observed during adipocyte differentiation or in mouse 3T3-L1 adipocytes [7,18,19], in fully differentiated human adipocytes, the expression of CTH was much lower than CBS and MPST.

No visual differences in lipid droplets were observed between control and combined partial knockdown cells (data not shown). Considering that adipocyte dedifferentiation becomes morphologically evident at least one week after the start of the process [20], longer gene knockdown experiments should be required to appreciate significant changes in intracellular lipid accumulation.

### 3.2. Impact of the Combined Partial Knockdown of MPST and CBS Genes on Adipocyte Protein Persulfidation

The possible link between CTH, CBS, and MPST activities and protein persulfidation during human adipocyte differentiation [8] suggests that these enzymes might modulate persulfidation in several adipogenic proteins, affecting the physiology of adipocytes. To test this hypothesis, we examined whether the reduction of adipogenic markers observed in human adipocytes with the combined partial knockdown of *MPST* and *CBS* genes was linked to altered persulfidation in key adipogenic proteins. For this purpose, we used a mass spectrometry label-free quantitative proteomic approach combined with the dimedone-switch method to measure the protein profile of persulfidation. Proteins from three biological replicates from fully differentiated human adipocytes, control, and the combined partial *MPST* and *CBS* gene knockdown were isolated and subjected to the dimedone-switch procedure for isolation of the persulfidated proteins. The proteomic analysis quantified 1834 proteins with a FDR threshold of 1% (Appendix A) of which 216 showed statistically different levels of persulfidation in KD cells compared to control adipocytes (Student *t*-test, *p* < 0.05) (Appendix A). From these proteins, 136 proteins were less persulfidated in KD cells, and 80 proteins were more persulfidated.

When functional protein association networks were explored using the STRING database, we found that those differentially less persulfidated proteins in human adipocytes with the combined partial knockdown of *MPST* and *CBS* genes were associated with relevant metabolic processes for adipocytes. These processes included the biosynthetic process of acetyl-CoA and acyl-CoA, tricarboxylic acid cycle, the metabolic process of pyruvate and glucose, extracellular matrix organization, the regulation of the mRNA stability, metabolic process of sulfur compound, the oxidation-reduction process, and the regulation of catabolic process (FDR < 0.02, Figure 2A). Otherwise, differentially increased persulfidated proteins in human adipocytes with the combined partial knockdown of *MPST* and *CBS* genes were mainly associated with processes related to immune response- and cytokine-mediated signaling pathways (FDR < 0.005, Figure 2B).

Specifically, among those proteins found less persulfidated, several relevant proteins for the maintenance of adipocyte adipogenesis and function were identified, including RAB4A, IDH2, ITGA5, NPC1, ACO1, LRP1, AHNAK, ARPC3, ADH1B, ALDH1A3, LGALS1, PHB, COL6A3, PLIN3, FABP4, THBS1, ANXA2, PLIN4, ACSL1, ANXA1, ELAVL1, ALDH1A1, and HNRNPA1 (Figure 3A, all with *p* < 0.05). Previous studies supported the relevance of these proteins in adipocyte biology. RAB4A is a small GTPase that participates in adipocyte GLUT4 trafficking, exerting an important role in glucose uptake [21,22]. IDH2 attenuates adipocyte inflammation through the induction of α-ketoglutarate production [23] and is required for lipogenesis in these cells [24]. ITGA5 promotes fibrosis of adipose tissue [25] and inhibits adipocyte differentiation in human adipose tissue-derived stem cells [26]. Decreased expression of the NPC1 gene or the partial deletion of this gene promoted weight gain and fat mass expansion [27,28,29], suggesting a relevant role of NPC1 in the prevention of obesity and adipocyte hypertrophy, possibly increasing lipolysis and beta oxidation and decreasing lipogenesis and triglyceride accumulation in adipocytes [30]. NPC1 is increased in adipocytes [31], but it was not associated with adipogenesis [32]. ACO1 is important to maintain adipogenesis in adipocytes, possibly modulating cytosolic NADPH levels and intracellular iron homeostasis, both processes required for adipocyte biology [33]. LRP1 is required for adipocyte differentiation and to maintain adipocyte adipogenesis [34], possibly in part due to its interaction with ShcA [35], or acting as APOA4 receptor, and favoring APOA4-induced glucose uptake [36]. Several studies have demonstrated that AHNAK is required for adipocyte differentiation [37,38,39]. Mechanistically, a recent study demonstrated that AHNAK gene knockdown decreased adipogenesis through the suppression of Bmpr1α expression and decreased BMP4/ Bmpr1α signaling [39]. ARPC3 is another protein required for adipocyte differentiation, exerting a relevant role in the late stage of the process, participating in GLUT4 exocytosis and insulin signal transduction, and suggesting a relevant role in maintaining insulin action in adipocytes [40]. ADH1B modulates intracellular lipid accumulation, without impacting adipogenesis [41], and preserved insulin-stimulated glucose uptake in adipocytes [42]. Inhibition of ALDH1A1 reduced visceral fat [43], negatively impacting adipogenesis in visceral adipose tissue [44]. LGALS1 (galectin 1) is an adipocyte secreted factor [45] that enhances the transcriptional activity of PPARG, increasing adipogenesis and accumulation of lipids in adipocytes [46]. Gain–loss in vivo studies demonstrated that this protein is crucial for adipose tissue expansion and adiposity [46]. LGALS1 expression in subcutaneous adipose tissue has been identified as a relevant factor to confer the risk of weight regain [47]. Pharmacological inhibition of LGALS1 displayed important anti-obesogenic and anti-adipogenic effects [48,49], strengthening the role of this protein in the development of fat mass and obesity. Otherwise, circulating galectin-1 levels were negatively associated with type 2 diabetes [50,51], indicating that it might be associated with healthy adiposity. PHB is induced during adipocyte differentiation, in response to insulin and PPARγ agonist, and its overexpression in 3T3-L1 fibroblasts was sufficient to induce adipogenesis [52], but its gene knockdown reduces the expression of adipogenic genes and lipid accumulation and impairs mitochondrial dysfunction [53,54]. In adipocytes, prohibitin is required for fatty acid uptake [55]. *COL6A3* gene knockdown increased the triglyceride content, lipolysis, insulin-induced Akt phosphorylation, and the expression of adipogenic genes (peroxisome proliferator-activated receptor-γ, glucose transporter, adiponectin, and fatty acid binding protein), indicating improved adipocyte function and insulin sensitivity [56]. In addition, COL6A3 knockdown also decreased basal adipocyte MCP1 mRNA expression, reduced secreted protein levels, and attenuated TNFα- and LPS-induced MCP1 gene expression [57]. In contrast, increased COL6A3 mRNA in adipocytes is associated with obesity, insulin resistance, and inflammation of adipose tissue [58,59], supporting the negative impact of this protein on adipocytes and adipose tissue. Annexin A1 (AnxA1) is an endogenous glucocorticoid regulated protein that modulates systemic anti-inflammatory processes. AnxA1 gene expression and protein were significantly up-regulated during adipogenesis in human SGBS preadipocytes [60]. Epididymal fat mass was reduced by ANXA1 gene deletion, but adipocyte size was unchanged, suggesting that ANXA1 is required for maintaining adipocyte cell number [61]. Furthermore, the progressive accumulation of Annexin A2 (AnxA2) in the myofiber matrix causes muscle-resident fibro/adipogenic precursors (FAP) differentiation into adipocytes, and depletion of AnxA2 prevents FAP adipogenesis and muscle loss [62]. ACSL1 is required for beta oxidation in adipocytes, and it is associated to adipose insulin sensitivity, adipogenesis and adipocyte fatty acid uptake [63,64,65,66,67,68,69,70]. ELAVL1 (also named HuR) is an important repressor of adipogenesis. Knockdown and overexpression of HuR in primary adipocyte culture enhances and inhibits adipogenesis, respectively [71]. HuR also exerts an important role in the prevention of adipocyte hypertrophy and inflammation of adipose tissue [72]. However, previous studies in 3T3-L1 cells indicate that HuR is required in early events of adipogenesis, allowing C/EBPβ mRNA translocation into the cytosol, and its proper translation, a process required to activate CEBPα and PPARγ in mitotic clonal expansion [73,74]. Importantly, the reduction in HuR persulfidation, resulted in the stabilization of HuR-target mRNAs, and increased its translation and expression [75], suggesting that the decrease in HuR persulfidation improved HuR dimerization and its activity, and consequently, it has negative effects on adipocyte adipogenesis. Interestingly, perilipin persulfidation inhibits lipid mobilization and lipolysis in adipocytes, and contributes to adequate maintenance of lipid storage [76]. HNRNPA1 is a known regulator of INSR exon 11 splicing, which could have an impact on adipose insulin action [77].

Among the most persulfidated proteins, some key mediators of adipocyte inflammation and dysfunction (IFI16, STAT3, RELA, STAT1, MAP4K4, PREB) were found, but also proteins that might have a positive role in adipogenesis (EXOC7, SLC39A14) (Figure 3B, all with *p* < 0.05). In mice and humans, increased IFI16 levels are associated with larger adipocytes, enhanced inflammatory state, and impaired insulin-stimulated glucose uptake in white adipose tissue [78]. STAT3 is a relevant transcription factor of proinflammatory genes in adipocytes that is associated with adipocyte inflammation [79,80]. RELA is a relevant transcription factor associated with adipocyte inflammation [81] that inhibits adipogenesis and accumulation of lipids in adipocytes [82]. The activation of STAT1 transcription factor inhibits adipogenesis and promotes adipocyte and adipose tissue inflammation [83,84,85,86,87,88]. Isolated mature adipocytes from obese individuals had increased expression of mitogen-activated protein 4 kinase 4 (MAP4K4), which is known to inhibit PPARγ transcriptional activity, adipogenesis, and insulin-stimulated glucose transport [89,90]. In line with this, MAP4K4 inhibits adipose lipogenesis via the suppression of Srebp-1 [91], and deletion of this kinase increases insulin sensitivity in adipose tissue [92]. Prolactin regulatory element-binding (PREB) is a negative regulator of adiponectin gene expression in adipocytes [93].

On the other hand, EXOC7 might participate in Glut4 trafficking [94,95], and SLC39A14 is a zinc transporter, which is rapidly induced in the early stages of differentiation, suggesting a possible role in adipogenesis [96]. In fact, depletion of SLC39A14 caused hypertrophy and inflammation of adipocytes [97].

These data indicated that the combined partial knockdown of the MPST and CBS genes disrupts the pattern of protein persulfidation in human adipocytes, suggesting a possible link between this disruption and adipocyte dysfunction, which is characterized by decreased adipogenesis but increased inflammation. This disruption included a significant decrease in persulfidation in key proteins for adipocyte function (RAB4A, IDH2, NPC1, ACO1, LRP1, AHNAK, ARPC3, ADH1B, ALDH1A1, LGALS1, PHB, ANXA1, ANXA2, ACSL1, PLIN3, PLIN4, FABP4, HNRNPA1), but also in proteins that negatively impact adipocytes and adipogenesis (ITGA5, COL6A3, ELAVL1). In addition, the combined partial knockdown of MPST and CBS genes in adipocytes resulted in increased persulfidation in relevant proinflammatory transcription factors (RELA, STAT1, STAT3) and anti-adipogenic proteins (IFI16, MAP4K4, PREB), but also in some proteins that might exert positive effects on adipogenesis, such as SLC39A14 and EXOC7. Persulfidation increases the reactivity of target proteins, modulating their biological activities [10]. However, it is important to note that modulation in persulfidation-induced protein activity can trigger activation or inhibition of the biological function of the persulfidated protein [98]. Other consequences of *CBS* and *MPST* gene knockdown, such as decreased H_2_S biosynthesis [8,18] or increased homocysteine levels [99], could also explain the negative impact on adipocyte function. Additional experiments using specific chemical inhibitors of H₂S biosynthesis and the administration of H_2_S donor molecules to CBS/MPST gene knockdown cells should be performed to further investigate the possible role of H_2_S in this model.

Selenium-binding protein 1 (SELENBP1), a fourth enzyme that participates in H_2_S biosynthesis, increased during differentiation of 3T3L1 adipocyte in association with lipogenesis and lipid accumulation, and it has been recently described as a marker of differentiated adipocytes [100]. *SELENBP1* gene knockdown had a negative impact on adipocyte function in parallel to the decreased expression of the CBS, CTH, and MPST enzymes [101]. The contribution of other enzymes, such as SELENBP1 [100,101], or non-enzymatic pathways [102] in H_2_S production cannot be excluded in the current study.

## 4. Conclusions

In summary, these findings indicate that the combined partial knockdown of CBS and MPST (but not CTH) in adipocytes impairs the expression of genes related with the maintenance of adipocyte function and promotes inflammation, possibly by disrupting the pattern of protein persulfidation in these cells. The current study points to the fact that these enzymes were not only required during adipocyte differentiation [8,9], but they were also needed for the functional maintenance of adipocytes.

## Figures and Tables

**Figure 1 antioxidants-11-01095-f001:**
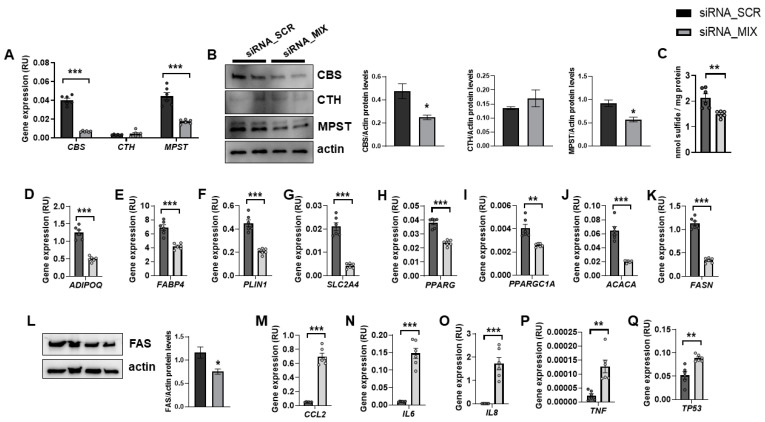
The impact of the combined partial knockdown of *CBS* and *MPST* gene on relevant genes for adipocyte biology. The effect of siRNA_MIX administration on *CBS*, *CTH* and *MPST* mRNA (**A**) and protein (**B**) levels, on intracellular sulfide levels (**C**), and on expression of adipogenic (*ADIPOQ, FABP4, PLIN1, SLC2A4, PPARG, PPARGC1A*)-, lipogenic (*ACACA, FASN*)-, proinflammatory (*CCL2, IL6, IL8, TNF*)- and cellular senescence (*TP53*)-related genes and on FAS protein levels (**D**–**Q**). * *p* < 0.05, ** *p* < 0.01 and *** *p* < 0.001 compared to siRNA_SCR.

**Figure 2 antioxidants-11-01095-f002:**
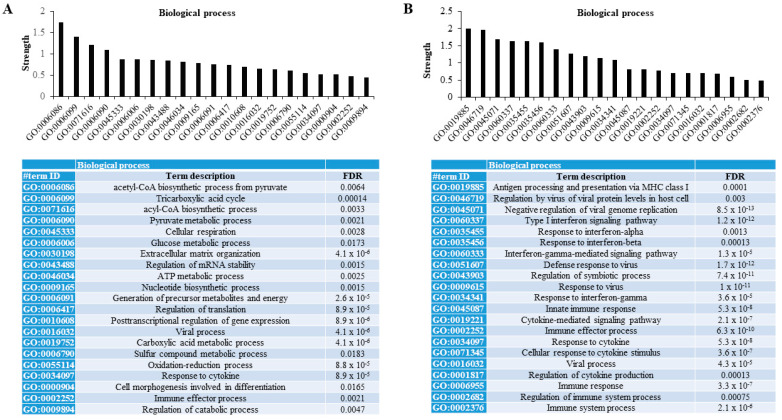
Proteomic persulfidation networks analysis. Analysis of functional protein association networks using STRING database of differentially decreased (**A**) and increased (**B**) persulfidated proteins in siRNA_MIX- compared to siRNA_SCR-treated cells. FDR, false discovery rate adjusted *p*-value.

**Figure 3 antioxidants-11-01095-f003:**
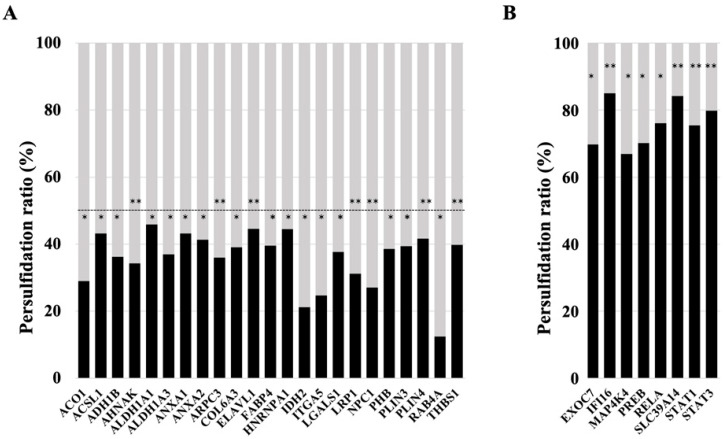
Rate of persulfidation of relevant proteins involved in adipocyte function and the maintenance of adipogenesis. Level of protein persulfidation in siRNA_MIX samples (black boxes, in percentage) compare to the level in control samples (grey boxes). (**A**) ACO1, Cytoplasmic aconitate hydratase; ACSL1, Long-chain-fatty-acid—CoA ligase 1; ADH1B, All-trans-retinol dehydrogenase [NAD(+)] ADH1B; AHNAK, Neuroblast differentiation-associated protein AHNAK; ALDH1A1, Retinal dehydrogenase 1; ALDH1A3, Aldehyde dehydrogenase family 1 member A3; ANXA1, Annexin A1; ANXA2, Annexin A2; ARPC3, Actin-related protein 2/3 complex subunit 3; COL6A3, Collagen alpha-3(VI) chain; ELAVL1, ELAV-like protein 1; FABP4, Fatty acid-binding protein adipocyte; HNRNPA1, Heterogeneous nuclear ribonucleoprotein A1; IDH2, Isocitrate dehydrogenase [NADP] mitochondrial; ITGA5, Integrin alpha-5; LGALS1, Galectin-1; LRP1, Low-density lipoprotein receptor-related protein 1; NPC1, NPC intracellular cholesterol transporter 1; PHB, Prohibitin; PLIN3, Perilipin-3; PLIN4, Perilipin-4;RAB4A, Ras-related protein Rab-4A; THBS1, Thrombospondin-1. (**B**) EXOC7, Exocyst complex component 7; IFI16, Gamma-interferon-inducible protein 16; MAP4K4, Mitogen-activated protein kinase kinase kinase kinase 4; PREB, Prolactin regulatory element-binding protein; RELA, Transcription factor p65; SLC39A14, Metal cation symporter ZIP14; STAT1, Signal transducer and activator of transcription 1-alpha/beta; STAT3, Signal transducer and activator of transcription 3. Student *t*-test significance: * *p* < 0.05; ** *p* < 0.01.

## Data Availability

Data are available via ProteomeXchange with identifier PXD032370.

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
