# Peer review of "The Combined Partial Knockdown of CBS and MPST Genes Induces Inflammation, Impairs Adipocyte Function-Related Gene Expression and Disrupts Protein Persulfidation in Human Adipocytes"

_antioxidants, 2022, doi:10.3390/antiox11061095_

Round 1
Reviewer 1 Report
The manuscript by Latorre et al is an nice paper and contains some novel findings. However, the work would be strengthened by the incorporation of some additional experimental data to confirm the functional role of H2S in this model. My comments are below.
1. Please read through the manuscript carefully and address any grammatical errors.
2. Could the authors use an alternate phrase rather than ‘joint knockdown’ throughout the paper. This causes confusion since the title implies the knockdown of CBS/3-MST in joint tissues. I suggest, ‘co-knockout’ or ‘combined knockout down’.
3. Figure 1. Can the authors explain why CBS/3-MST protein levels are only approx. 50% reduced in adipocyte tissue. The current results indicate that only a partial knockdown of the targets genes/proteins has been achieved. Therefore, it may be more appropriate to use the phrase ‘partial knockdown’.
4. Can the authors provide results associated with lipid content of adipocytes using oil-red staining? This to indicate changes in lipid levels in adipocytes and the impact of gene knockouts? Glycerol levels could also be assessed.
5. The results would be strengthened be the incorporation of enzyme activity data showing reduced production of H2S by the target enzymes in cells.
6. Inhibitor studies should be used in the current work targeting both CBS/3-MST. Inhibitors should cause parallel changes in the expression of the selected target genes and confirm the importance of these proteins and H2S in this system.
7. Knockdown of CBS/3-MST should reduce the levels of several adipogenic markers and promote changes in the lipid content of cells. Can changes in these markers be reverse by the use of H2S donor molecules? This series of minor experiments would confirm a role of H2S in this model
Author Response
Manuscript Number: antioxidants-1681092 formerly entitled “The joint knockdown of CBS and MPST genes induces inflammation, impairs adipogenesis and disrupts protein persulfidation in human adipocytes” and now entitled:
“The combined partial knockdown of CBS and MPST genes induces inflammation, impairs adipocyte function-related gene expression and disrupts protein persulfidation in human adipocytes”
The authors are grateful for the reviewers’ and editor comments which have contributed to clarify the message of our paper and to improve the quality of our submission. Please note that we have been given a very short time frame to provide our responses, so additional experiments have not been possible to perform. Many thanks for your understanding. The specific comments are addressed below:
Reviewer 1
Comments and Suggestions for Authors
The manuscript by Latorre et al is an nice paper and contains some novel findings. However, the work would be strengthened by the incorporation of some additional experimental data to confirm the functional role of H2S in this model. My comments are below.
We thank the reviewer for the positive view of our manuscript and the excellent comments and suggestions. A detailed point-by-point response to the comments is included below.
- Please read through the manuscript carefully and address any grammatical errors.
The manuscript has been now carefully revised and grammatical errors addressed accordingly.
- Could the authors use an alternate phrase rather than ‘joint knockdown’ throughout the paper. This causes confusion since the title implies the knockdown of CBS/3-MST in joint tissues. I suggest, ‘co-knockout’ or ‘combined knockout down’.
Thank you. To clarify this point “joint knockdown” is now changed into “combined partial knockdown”.
- Figure 1. Can the authors explain why CBS/3-MST protein levels are only approx. 50% reduced in adipocyte tissue. The current results indicate that only a partial knockdown of the targets genes/proteins has been achieved. Therefore, it may be more appropriate to use the phrase ‘partial knockdown’.
We agree with the reviewer. The term “knockdown” has now been changed into “partial knockdown”.
- Can the authors provide results associated with lipid content of adipocytes using oil-red staining? This to indicate changes in lipid levels in adipocytes and the impact of gene knockouts? Glycerol levels could also be assessed.
The reviewer is right with this appreciation. We fully agree the fact that changes in lipid and glycerol levels could further demonstrate the impact of gene knockdowns on adipocyte biology. Since these experiments were designed to evaluate RNA and protein levels, no samples were collected to analyze lipid or glycerol levels. Nevertheless, no visual differences in lipid droplets between control and combined partial knockdown cells were observed at the end of the experiment. However, it is important to note that current experiments lasted 72 h, and considering that adipocyte dedifferentiation becomes morphologically evident at least one week after the start of the process (PMID: 3732657), longer experiments may be necessary to appreciate significant changes in intracellular fat accumulation. The following text has now been included in results and discussion section:
“No visual differences in lipid droplets between control and combined partial knockdown cells were observed (data not shown). Considering that adipocyte dedifferentiation becomes morphologically evident at least one week after the start of the process [19], longer gene knockdown experiments should be required to appreciate significant changes in intracellular lipid accumulation.”
- Sugihara H, Yonemitsu N, Miyabara S, Yun K (1986) Primary cultures of unilocular fat cells: characteristics of growth in vitro and changes in differentiation properties. Differentiation 31(1):42–49. https://doi.org/10.1111/j.1432-0436.1986.tb00381.x
- The results would be strengthened be the incorporation of enzyme activity data showing reduced production of H2S by the target enzymes in cells.
We agree with the reviewer suggestion. Since the impact of these partial knockdowns on H2S production has to be analyzed in live cells, the absence of these enzyme activity data is a limitation of current study that needs to be addressed in future experiments. The following sentence in the last paragraph of the discussion has now been added to acknowledge this limitation:
“Other consequences of CBS and MPST gene knockdown, such as the decrease in H2S biosynthesis[8,17,97] or the increase in homocysteine levels[97], might also explain the negative impact on adipocyte adipogenesis. These should be assayed in future studies.”
- Inhibitor studies should be used in the current work targeting both CBS/3-MST. Inhibitors should cause parallel changes in the expression of the selected target genes and confirm the importance of these proteins and H2S in this system.
Thank you very much for this suggestion. The following text is now included in discussion:
Additional experiments using specific chemical inhibitors of Hâ‚‚S biosynthesis and performing administration of H2S donor molecules in CBS/MPST gene knockdown cells should be performed to further investigate the possible role of H2S in this model.
- Knockdown of CBS/3-MST should reduce the levels of several adipogenic markers and promote changes in the lipid content of cells. Can changes in these markers be reverse by the use of H2S donor molecules? This series of minor experiments would confirm a role of H2S in this model
This is a very interesting suggestion that requires further experiments. A new text has now been included in discussion to address this limitation. Please see above response.
Reviewer 2 Report
Hydrogen sulfide is increasingly being recognized as an important signaling molecule, affecting the function and fate of many types of cells. In this regard, the present manuscript continues previous studies on H2S and H2S-producing enzymes in adipocytes. The authors provide experimental evidence that an siRNA-mediated joint knockdown of CBS and MPST results in the down-regulation of a number of genes/proteins, which are crucial for adipocyte function, whereas pro-inflammatory genes are up-regulated. In the second part of the paper, the authors provide an overview on alterations in protein persulfidation in response to CBS/MPST-knockdown, which is expected to stimulate further research in the field. To my mind, this is a small but interesting and well-performed paper. I have only two comments/suggestions on the interpretation and discussion of the data.
1) To my mind, one of the conclusions ("CBS/MPST-knockdown impairs adipogenesis") cannot be drawn from the data provided here and should be deleted throughout the text and also the title. As the authors transfected cells with anti-CBS/MPST siRNA that were already terminally differentiated adipocytes, they did not explore the effect of CBS and MPST knockdown on the process of adipocyte differentiation. While the genes explored in this context are indeed induced during adipocyte differentiation they are also involved in the maintenance of adipocyte function. If the authors intend to proove an effect of the applied CBS/MPST-siRNA on adipogenesis, they will have to treat preadipocytes as well and check also some genes (such as C/EBP-beta) that are characteristic for the early adipocyte differentiation stage.
2) Two recent studies (PMID: 30469030; PMID: 33673622) have demonstrated the presence and functional relevance of a fourth H2S-producing enzyme, SELENBP1, in adipocytes. These studies should be discussed as well by the authors. Most interestingly, SELENBP1 appears to be highly expressed in mature adipocytes, and its knockdown was demonstrated to downregulate both CBS and MPST as well as to impair adipogenesis.
Author Response
Manuscript Number: antioxidants-1681092 formerly entitled “The joint knockdown of CBS and MPST genes induces inflammation, impairs adipogenesis and disrupts protein persulfidation in human adipocytes” and now entitled:
“The combined partial knockdown of CBS and MPST genes induces inflammation, impairs adipocyte function-related gene expression and disrupts protein persulfidation in human adipocytes”
The authors are grateful for the reviewers’ and editor comments which have contributed to clarify the message of our paper and to improve the quality of our submission. Please note that we have been given a very short time frame to provide our responses, so additional experiments have not been possible to perform. Many thanks for your understanding. The specific comments are addressed below:
Reviewer 2
Comments and Suggestions for Authors
Hydrogen sulfide is increasingly being recognized as an important signaling molecule, affecting the function and fate of many types of cells. In this regard, the present manuscript continues previous studies on H2S and H2S-producing enzymes in adipocytes. The authors provide experimental evidence that an siRNA-mediated joint knockdown of CBS and MPST results in the down-regulation of a number of genes/proteins, which are crucial for adipocyte function, whereas pro-inflammatory genes are up-regulated. In the second part of the paper, the authors provide an overview on alterations in protein persulfidation in response to CBS/MPST-knockdown, which is expected to stimulate further research in the field. To my mind, this is a small but interesting and well-performed paper. I have only two comments/suggestions on the interpretation and discussion of the data.
We thank the reviewer for the positive view of our manuscript and the excellent comments and suggestions. A detailed point-by-point response to the comments is included below.
1) To my mind, one of the conclusions ("CBS/MPST-knockdown impairs adipogenesis") cannot be drawn from the data provided here and should be deleted throughout the text and also the title. As the authors transfected cells with anti-CBS/MPST siRNA that were already terminally differentiated adipocytes, they did not explore the effect of CBS and MPST knockdown on the process of adipocyte differentiation. While the genes explored in this context are indeed induced during adipocyte differentiation they are also involved in the maintenance of adipocyte function. If the authors intend to prove an effect of the applied CBS/MPST-siRNA on adipogenesis, they will have to treat preadipocytes as well and check also some genes (such as C/EBP-beta) that are characteristic for the early adipocyte differentiation stage.
Thank you for this appreciation. We fully agree with the reviewer. We have now changed “impairs adipogenesis” by “impairs expression of the maintenance of adipocyte function-related genes”. Please see below:
In title:
The combined partial knockdown of CBS and MPST genes induces inflammation, impairs adipocyte function-related gene expression and disrupts protein persulfidation in human adipocytes
In abstract:
In conclusion, current study indicates that joint knockdown of CBS and MPST (but not CTH) in adipocytes impairs expression of the maintenance of adipocyte function-related genes and promotes inflammation, …
In conclusions:
To sum up, these findings indicate that joint knockdown of CBS and MPST (but not CTH) in adipocytes impairs expression of the maintenance of adipocyte function-related genes and promotes inflammation, …
2) Two recent studies (PMID: 30469030; PMID: 33673622) have demonstrated the presence and functional relevance of a fourth H2S-producing enzyme, SELENBP1, in adipocytes. These studies should be discussed as well by the authors. Most interestingly, SELENBP1 appears to be highly expressed in mature adipocytes, and its knockdown was demonstrated to downregulate both CBS and MPST as well as to impair adipogenesis.
Thank you for this interesting suggestion. The following text has now been added in discussion:
Selenium-binding protein 1 (SELENBP1), a fourth enzyme that participate in H2S biosynthesis, increased during in 3T3L1 adipocyte differentiation in association with lipogenesis and lipid accumulation and it has been recently described as a marker of differentiated adipocytes [99]. SELENBP1 gene knockdown had a negative impact on adipocyte function in parallel to decreased expression of CBS, CTH and MPST enzymes [100].
- Steinbrenner H, Micoogullari M, Hoang NA, Bergheim I, Klotz L-O, Sies H (2019) Selenium-binding protein 1 (SELENBP1) is a marker of mature adipocytes. Redox Biol 20:489–495. https://doi.org/10.1016/j.redox.2018.11.004
- Randi EB, Casili G, Jacquemai S, Szabo C (2021) Selenium-Binding Protein 1 (SELENBP1) Supports Hydrogen Sulfide Biosynthesis and Adipogenesis. Antioxidants (Basel, Switzerland) 10(3). https://doi.org/10.3390/antiox10030361
Round 2
Reviewer 1 Report
No specific comments. I still feel that the work would be strengthened by the inclusion of the additional experiments. These are short experiments but add the necessary controls to confirm the role of the target enzymes in the processes described in adipocytes.
Author Response
Manuscript Number: antioxidants-1681092 entitled “The combined partial knockdown of CBS and MPST genes induces inflammation, impairs adipocyte function-related gene expression and disrupts protein persulfidation in human adipocytes”
The authors are grateful for the reviewer’s comments which have contributed to clarify the message of our paper and to improve the quality of our submission. The specific comments are addressed below:
Reviewer 1
Comments and Suggestions for Authors
No specific comments. I still feel that the work would be strengthened by the inclusion of the additional experiments. These are short experiments but add the necessary controls to confirm the role of the target enzymes in the processes described in adipocytes.
Thank you. I assume you are referring to those experiments you mentioned in the first review round:
- The results would be strengthened be the incorporation of enzyme activity data showing reduced production of H2S by the target enzymes in cells.
- Inhibitor studies should be used in the current work targeting both CBS/3-MST. Inhibitors should cause parallel changes in the expression of the selected target genes and confirm the importance of these proteins and H2S in this system.
- Knockdown of CBS/3-MST should reduce the levels of several adipogenic markers and promote changes in the lipid content of cells. Can changes in these markers be reverse by the use of H2S donor molecules? This series of minor experiments would confirm a role of H2S in this model
We agree in which these experiments are required to demonstrate if combined partial knockdown of CBS and MPST genes attenuates H2S biosynthesis, and to investigate the possible role of H2S in this model. However, this is beyond the scope of this study. In addition, it cannot be excluded that H2S production could be compensate by other enzymes, such as Selenium-binding protein 1 (SELENBP1) [99,100], or by non-enzymatic pathways [101].
To include this information, the following text has now been added into discussion:
The contribution of other enzymes, such as SELENBP1 [99,100], or non-enzymatic pathways [101] in H2S production cannot be excluded in current study.
- Yang J, Minkler P, Grove D, Wang R, Willard B, Dweik R, Hine C (2019) Non-enzymatichydrogen sulfide production from cysteine in blood is catalyzed by iron and vitamin B6. Commun Biol 2(194). https:// doi: 10.1038/s42003-019-0431-5.